# Olfactory Strategies in the Defensive Behaviour of Insects

**DOI:** 10.3390/insects13050470

**Published:** 2022-05-18

**Authors:** Kavitha Kannan, C. Giovanni Galizia, Morgane Nouvian

**Affiliations:** 1Department of Biology, University of Konstanz, 78457 Konstanz, Germany; giovanni.galizia@uni-konstanz.de; 2Centre for the Advanced Study of Collective Behaviour, University of Konstanz, 78457 Konstanz, Germany; 3Zukunftskolleg, University of Konstanz, 78457 Konstanz, Germany

**Keywords:** chemical defense, alarm pheromone, olfactory strategies, odorant coding, aggression, defensive behaviour

## Abstract

**Simple Summary:**

Insects have several methods to protect themselves and their resources from danger. One of them is to use their sense of smell. In this review, we describe how insects use smell to detect threats and perform behaviours of ‘flight or fight’ such as avoidance, escape or attack, in order to protect themselves. We also discuss how group-living insects share the information of danger through semiochemicals called alarm pheromones, to act as a collective. In the second section of this paper, we review how these odours are processed in insect brains. We discuss how the two kinds of neural architectures observed in olfactory areas, labelled-lines and across-fiber patterns, support the processing of alarm pheromones. Finally, we give an outlook on potential future studies that will help us understand this field better.

**Abstract:**

Most animals must defend themselves in order to survive. Defensive behaviour includes detecting predators or intruders, avoiding them by staying low-key or escaping or deterring them away by means of aggressive behaviour, i.e., attacking them. Responses vary across insect species, ranging from individual responses to coordinated group attacks in group-living species. Among different modalities of sensory perception, insects predominantly use the sense of smell to detect predators, intruders, and other threats. Furthermore, social insects, such as honeybees and ants, communicate about danger by means of alarm pheromones. In this review, we focus on how olfaction is put to use by insects in defensive behaviour. We review the knowledge of how chemical signals such as the alarm pheromone are processed in the insect brain. We further discuss future studies for understanding defensive behaviour and the role of olfaction.

## 1. Introduction

Animals living in the same environment often face intense competition with each other for basic needs like food, mate, shelter, and territory. Members of the same and different species fight for these assets, and thus, protecting them becomes essential for every individual animal. Sometimes, an individual is regarded as prey, and in such cases, it needs to protect itself from the predator. Apart from protecting oneself and one’s assets, individuals belonging to social groups protect their conspecifics from intruding and predating animals, by communicating danger signals and/or displaying aggressive behaviour to deter them away. Together, these acts of protecting oneself, along with the protection of their kin and the assets, can be categorised under an important behaviour common to most animal species, which is called ‘defensive behaviour’.

In ecology, the study of defensive behaviour is fundamental to understanding how an individual animal survives, given the multiple factors of the threat it encounters. For insects, threats range from an intruding neighbour (i.e., a conspecific) competing for food resources, to attacks on individuals (by predators or parasitoids) to an attack on the entire group or colony from other insects or larger vertebrates. Most animals, including insects, employ their senses of vision, olfaction, and audition among others, to detect danger, communicate it and display defensive responses. 

In this review, we aim to understand how insects use their sense of smell in defensive behaviour. We divide the review into two sections, the first focusing on behaviour and the second on neurobiology. In the first, we describe how different insects use odours to detect, signal and/or communicate through different modes of defensive behaviours. In the second, we describe how these odours are processed as information in the brain. Finally, we discuss open questions and necessary future studies to understand the role that olfaction plays in the defensive behaviour of insects.

## 2. Olfaction in Defensive Behaviours of Insects

Different species of insects use different strategies in defending themselves from an intruder or predator. In this section, we outline how some insect individuals begin to defend directly after olfactorily detecting danger cues, while some others then communicate it as olfactory signals to their conspecifics before starting to defend (see Figure 1). We address questions of what the differences in defensive behaviour are, by giving examples from various insect species (both solitary and group-living). We also focus on the contrasting olfactory interactions of an individual insect with others, e.g., with a predator, an intruder, a heterospecific or conspecifics of the same and different groups, in threatening situations. The different olfactory strategies of insects are described in detail (Figure 1). 

## 3. Defensive Behaviour Based on Olfactory Detection

The first step in the defensive behaviour is to detect the threat. Animals use all their senses, including olfaction, to manage the costs and risks of predation, intrusion, and danger [1]. In the case of predation, being noticeable to the predator is a large risk for the prey. Thus, the first defensive response is to avoid the predator, by not entering the areas visited by the predator. One among several examples of insects that showcase this behaviour is the larvae of the Blue-tailed damselfly *Ischnura elegans*, which avoid profitable patches when foraging if they chemically detect the presence of their predator, a water beetle, *Notonecta glauca* [2]. The next defensive strategy that is often observed is, the alteration of one’s own behaviour upon olfactorily detecting the predator, to reduce the risk of attack [1,3]. This ranges from laying low by being immobile (freezing) or reducing activity to escaping from the dangerous zone. These different reactions may be observed in the same species, depending on the predator characteristics. Adult Tephritid fruit flies (*Bactrocera tryoni*) alter their behaviour differently depending on the predator their encounter [4]. As observed in an experimental assay, the flies reduced their movements by exhibiting a ‘freezing behaviour’ in the presence of a nocturnal predator odorants, but increased their movements by showing a ‘fleeing behaviour’ in the presence of diurnal predator odorants. It may be that diurnal predators mostly depend upon vision and can easily spot the fly even if it is immobile, such that fleeing is the only effective defensive strategy in this case. By contrast, nocturnal predators may rely more on vibratory cues, so reducing movements in their presence decreases the chances of being detected. Similar results were also observed in the responses of wood crickets (*Nemobius sylvestris*) to the odours of predatory spiders [5]. The larvae of fruit flies (*Drosophila melanogaster*) also provide an example of escaping behaviour: when they smell iridomyrmecin, the sex pheromone from the parasitoid wasps *Leptopilina* species, they react immediately to it by crawling away [6].

Apart from causing changes in motility and exploration, odorant cues from predators can also alter (mainly decrease) other fitness-associated activities such as foraging and mating [4,7,8]. As a response to the detection of odorants linked to defense, an insect may also shift its attention towards this challenging situation, at the detriment of its other activities. More specifically, honeybees (*Apis mellifera*) usually learn to associate a floral odour with a reward if both are presented simultaneously [9]. However, bees lose this ability if they have been exposed to iso-amyl acetate, a component of the honeybee alarm pheromone, prior to training [10]. Similarly, *Apis cerana* bees create an association between a feeding location and “neutral” odours, but do not do so when the odours used are representative of threats [11,12]. Interestingly, associative learning and memory retention of the task was better when using their alarm pheromone as a conditioned stimulus than the odour of a predatory hornet (*Vespa velutina*). This slight difference reflects how bees assess predation risk. The hornet odour represents a direct threat (high risk), while the sting alarm pheromone is part of an information channel about a threat (indirect risk). Detecting the hornet odour is a signal that it is present in the nearby surroundings, necessitating an immediate reaction. By contrast, the alarm pheromone is a broadcasted message carrying no specific information about the danger, and could be outdated. Finally, the presence of a predator in the environment may also induce behavioural adaptations on a longer timescale. For example, *Reticulitermes grassei* termites that have been exposed to the odour signature of a predatory ant (*Lasius niger*) for 2 months maintain a higher level of aggressiveness than their naïve counterparts [13].

Another interesting defensive strategy following olfactory detection and danger assessment is manipulating the predator’s own olfactory signals with external substances. This is observed with the example of the Asian honeybees, *Apis cerana,* and *Apis dorsata*, which undergo high predation risk from giant hornets, *Vespa mandarinia*. Hornet scouts identify vulnerable hives and mark them with their scent. Following this, the hornets recruit nestmates and return as groups to attack the hive by following their scent marks. Although the bees cannot fight back against the hornet group, they can cover up the markings from the hornet scouts. To do so, bees smear their nests with external substances, such as plant extracts [14] or faeces of vertebrate animals [15]. This defensive strategy of the bees deters the hornets as they can no longer find the hive, or are repelled by the foul smells. Similarly, a recent study showed that the alarm pheromone of aphids (E)-β-farnesene can alter the behaviour of its intraguild predators, the gallmidge larvae *Aphidoletes aphidimyza* and the anthocorid bug *Orius laevigatus* [16].

Following olfactory detection of the threat, vulnerable preys can also fight rather than hide or flee. The ‘fight’ or ‘aggressive behaviour’ is a set of counter-threatening behaviours that deters the attacker from further attacking. Most often, the animals that fight back are equipped with anti-predator adaptations [1] such as morphological (e.g., stinger, claw, horn, teeth) and physiological defenses (e.g., spraying chemicals, venom, speed of flight, aposematism). Terrestrial arthropods (insects, spiders, scorpions, centipedes) display forms of these defenses that have been studied in detail [17,18]. For example, Bombardier beetles *Stenaptinus insignis*, have a unique mechanism that gives them their name—they eject a bomb of a hot noxious spray of p-benzoquinones from their pygidial glands in their abdomen with a popping sound [19,20]. A species of stick insects in the United States, called the “Devil Rider” *Anisomorpha buprestoides*, spray terpenes from their pygidial glands that are strongly irritating to humans and other animals [21]. In the context of this review, it is interesting to note that some of these chemical weapons have also evolutionarily acquired a function in olfactory communication. Some members of the ant subfamily Dolichoderinae such as *Iridomyrmex pruinosus* and *Tapinoma* species release an alarm pheromone consisting of methyl-n-amyl-ketone from their pygidial gland that leaves a ‘rancid coconut odour’ in addition to eliciting a strong alarm behaviour in the recipient conspecifics [22]. Similarly, formicine ants spray formic acid against their intruder, which also serves as their alarm pheromone [23]. A recent study suggests that evolution can also take the reverse path: the stingless bee *Lestrimelitta niitkib*, which specializes in raiding other species, produces unusually high amounts of common mandibular alarm pheromone compounds. Such a high dose of these compounds, injected through biting, is toxic to bees of similar sizes, thus conferring a fighting advantage to *L. niitkib* [24].

Olfactory communication can happen not only between conspecifics, but also between the prey and its predator. Some insects release odours and chemicals that warn predators to not attack or consume them. This can be described as olfactory aposematism [25], where warning by odours is a conditioned stimulus of deleterious effects (e.g., bitter taste of chemicals and poison, physiological discomfort) that the predator has to learn. It is thus benefitting the species as a whole rather than the individual. The chemicals ejected by the stink bug (*Cosmopepla bimaculata)* upon mild tactile stimulation, is a classic example of olfactory aposematism [26,27,28]. The odour of the stink bug defensive secretion is so strong and aversive that a predator bird (Starling, *Sturnus vulgaris*), immediately ejected the still-living bug from its beak after its first attack, and ignored all subsequent stink bugs. In a similar fashion, the anole lizard *(Anolis carolensis)*, initially showed mild aversive behaviour upon feeding on the bugs, demonstrated by repeatedly wiping off its snout to remove the odour and taste. When tested on a consecutive day, the lizards showed stronger aversive and avoidance behaviour. The authors state that illness following consumption of the bugs could have enabled this stronger reaction towards them on a consecutive day [26]. Similarly, locusts (*Locusta migratoria*) in their gregarious form also produce an aposematic odour that is degraded into a toxic compound after ingestion [29]. Generally, these chemicals do not elicit any behavioural responses in the releasing insect itself; whether and which insects have olfactory receptors that are capable of detecting their own species’ chemical defense substances remains to be elucidated.

Threat detection includes an additional challenge for social insects because the threat can take the form of a conspecific from a different colony. This is especially frequent when there is competition for limited resources. Often, specialized individuals (guards) are placed at the nest entrance and are responsible for expelling non-nestmates attempting to enter the colony. Identification of non-nestmates relies on odour cues, in particular cuticular hydrocarbons (CHCs), and sometimes on other substances that individuals acquire by contact with their nest material, e.g., fatty acids and esters derived from the comb in the case of honeybees [30,31,32,33,34,35,36,37]. Interestingly, a study in carpenter ants *(Camponotus herculeanus)* has shown that guards do not specifically recognize the olfactory signature of nestmates, but rather recognize and reject individuals carrying odour cues novel to their own colony CHCs [38]. Nestmate recognition mechanisms are complex and have been reviewed extensively before, hence we invite our readers to refer to this work for further information [30,39,40,41,42]. 

## 4. Defensive Behaviour Based on Olfactory Communication

In the previous section, we described the differential defensive strategies (‘flight’ or ‘fight’) as direct responses to the threat, and here we describe how alarm pheromones elicit similar behaviours. While most insects need to rely on their own senses to detect a threat, in social species this information can be passed on to other group members, so that an appropriate response can be elicited at the collective level. This communication can take various forms including visual, acoustic, and of course olfactory signals. Visual and acoustic modalities of communication work efficiently, but necessitate constant signalling from the sender. Receivers may miss the message due to the non-binding properties of sound or light. Chemical signals, on the other hand, consist of molecules that stay in the atmosphere for various durations depending on their volatility. They also have specific binding properties to olfactory receptors which receive the signal and generate a neural signal (for more details; see next section on neurobiology). Moreover, there has been a strong selection towards sensitivity and specificity within chemical communication [43]. 

Specific chemical compounds known as ‘pheromones’ are used to convey a large number of messages between a sender and receiver animal of the same species [43,44]. Some pheromones consist of a single chemical component while others can be complex mixtures of chemicals, in which the relative concentrations of the components are important. Pheromones are chemical labels with either an innate or imprinted meaning, and thus they act as unique signals in the natural environment of a particular animal species. They are produced from different glands that are often located on body parts relevant to their function. Honeybees have at least 50 glands producing pheromonal compounds, and the community is still counting [45]. Pheromones range widely in function: sexual pheromones are used by females to attract males (or vice versa), aggregation pheromones are used to create leks, trail pheromones to mark trails and facilitate navigation, and alarm pheromones to communicate danger. They elicit reactions that are often stereotypical, but that can be modulated by the individual’s state or experience [46]. From an olfactory coding point of view (see next section on neurobiology for more details), pheromone identification is either innate or fixed early in life, and (short-term) learned plasticity should not modify its coding. On longer time scales, plasticity may adapt an animal to changing developmental or seasonal needs, e.g., when pheromones are only relevant in particular situations: sexually immature animals do not react to sexual pheromones, or–in the context of this review–guard bees may code alarm pheromone more efficiently than nurse bees (age polyethism). Pheromones can be classified as ‘releasers’ or ‘primers’ depending on the reaction they elicit, although some belong to both categories. Releasers trigger an immediate behavioural response while primers have a long-term physiological, modulatory, or biochemical effect [43,47]. 

In this review, we focus on ‘alarm pheromones’ that signal danger to other group members. They are known to occur in group-living species of the orders Hymenoptera, Isoptera, Homoptera, and Heteroptera [48]. While signalling the danger is beneficial to the group, it is worth noting that it comes at a cost for the individual sender, who has to produce the compounds and risks being more noticeable by the threat because of this signalling [49]. Not all social species possess an alarm pheromone, especially species with smaller colonies, such as bumblebees, some wasps (e.g., *Polistes dubia*), and some ants (e.g., *Myrmecina graminicola)* do not seem to have one [50]. Social insects are often equipped with organs of defense as anti-predator adaptations, such as the mandibles and the stinger apparatus, and most often, the alarm substances are linked to these organs (via mandibular and sting glands) [50]. This is mainly true for Hymenopterans such as honeybees (*Apis* species), wasps (*Vespa* species), and ants (*Pogonomyrmex* species, *Oecophylla* species, and *Formica* species, among several others) [50]. Alarm pheromones are mainly releasers, switching an individual immediately from any behavioural task into a ‘defense mode’. Additionally, they often mark the location of the disturbance [23,51]. Individual insects act immediately upon receiving the alarm signal with either a ‘fright and escape’ behaviour or a ‘recruit and act aggressive’ behaviour depending on the species [52]. For example, on the release of the aphid alarm pheromone (E)-β-farnesene, which is secreted and released from the cornicles situated on the abdomen of the aphid, these gregarious, group-living insects drop off their host plants in an escape response [52,53]. Similarly, the ants *Lasius alienus* are immediately triggered by their alarm substances. They run in an erratic and scattered pattern when they are a smaller colony, resulting in a “panic alarm response”. When worker ants form a larger and more compact colony, they still run erratically under the release of their alarm pheromone, however, most move deliberately towards alien objects and attack without hesitation, thus eliciting a “stand and hold defence”. This shift from panic to “aggressive alarm response” is often observed in Hymenopteran species as the colonies grow larger [23,54]. Alarm pheromones are thus either repellent when they trigger escape, or attractant when their function is to recruit more individuals to attack. Because of this, they are sometimes used as in the management of pests species (ants and aphids) in agricultural settings, either to push them away from a crop or to gather them into traps [52,55].

In addition to being a signal for conspecifics, alarm pheromones also allow different species to eavesdrop, sometimes to the benefit of the emitter species (commensalism) and sometimes to their detriment (parasitism and predation). *Apis cerana* foragers benefit by eavesdropping on the alarm pheromone signals of other *Apis* species i.e., *Apis dorsata* and *Apis mellifera*, and avoiding their predators after doing so [56,57]. This shows that alarm pheromone signalling is of use to members of the same species as well as different species, as public information contributes to avoiding shared predators. Wang et al. [56] showed that *Apis cerana* react to gamma-octanoic lactone (GOL), a component of the alarm pheromone that they do not produce, but is released by the co-occurring honeybee species *Apis dorsata*. Similarly, a recent study found that 7 out of 16 ant species tested were alerted upon smelling the alarm pheromone of the common canopy ant, *Azteca trigona* [58]. On the other hand, when information is public, it also draws unwanted attention. For example, the alarm pheromone of the fire ant *Solenopsis invicta* attracts parasitoid phorid flies *Pseudateon tricuspis* [59]. Moreover, it was shown in a recent study that many insects could detect and respond to the alarm pheromone component of this fire ant. This is of disadvantage to the ant because it is a predator species and the public information of its alarm information could deter its prey, allowing it to escape [60]. Likewise, *A. cerana* bees respond to the alarm pheromones of the hornets they co-evolved with by exhibiting appropriate defensive responses, whereas the allopatric *A. mellifera* do not show such behavioural adaptation [12,61]. 

## 5. The Sting and Mandibular Alarm Pheromones

The sting alarm pheromone (SAP) has been widely studied for its role in triggering an aggressive response. Although the SAP is prevalent in a large number of Hymenopteran species, most studies focus on honeybees. Beekeepers know well the banana-like scent that occurs upon opening or disturbing bee colonies, and that precedes an attack by the bees. Indeed when they detect the presence of a large threat, either by sight, smell or vibrations, honeybees release this alarm pheromone. It is composed of at least 40 different compounds, including isoamyl acetate (IAA), butyl acetate, 1-hexanol, and n-butanol, among several others [62,63]. It was later found that the SAP is produced from the Koschewnikov glands and the proximal parts of the sting sheath in the abdomen of the bee [64,65]. IAA triggers recruitment and attack both in the field and in the lab [66,67,68]. The attack involves biting, mauling, hair-pulling and stinging-which, in turn, releases more SAP and marks the predator [67,69]. We have described this sequence in Figure 2.

Some other species of Aculeata (i.e., Hymenopterans equipped with an anatomical modification of the stinger) such as wasps and hornets, have their alarm pheromone derived from the venom secreted during the stinging act [48,50,70]. This elicits a similar aggressive response as that of honeybees [71]. For example, venom extracted alarm compounds such as 2-pentanol and 3-methyl-1-butanol from giant hornets *Vespa mandarinia* cause excited flying, followed up by a further alarm, recruitment, and defensive behaviour such as stinging and biting [72]. 

Other than the sting alarm pheromone, many group-living insects (especially Hymenopterans and Isopterans such as honeybees, stingless bees, wasps, ants, and termites) use an alarm pheromone secreted from their mandibular glands. These glands are located on each side of the anterior head cavity and release the mandibular alarm pheromone (MAP) by means of an internal pore on the mandibles. The MAP often consists of volatile alcohols and ketones [48], for example, 4-methyl-3-heptanone in *Atta texana* ants [73] and 2-heptanone in honeybees [74,75]. Similar to the SAP, the MAP mostly elicits a ‘recruit and attack’ reaction. Not surprisingly, the release of MAP is most prominent in social insects that lack a stinger, such as many ants, stingless bees, and termites [23,48,76,77]. As described in the previous section, the MAP sometimes doubles as a chemical weapon, for example by acting as an irritant against vertebrate predators [78]. In insects that produce both SAP and MAP, it is often observed that they are each used in different situations. For example, honeybees release the mandibular alarm pheromone to defend against other insects (“small threat”) as compared to releasing the sting alarm pheromone to defend against larger vertebrate mammals (“large threat”) [67]. These pheromones sometimes also occur in tandem with the secretion of the same or different glands, thus exhibiting a combined attack force in their alarm-defense system. For example, in formicine ants, the workers release the mandibular alarm pheromone, along with the Dufour’s gland pheromone (decyl, dodecyl, tetradecyl acetate), as well as spray formic acid at the same time, to result in a ’mass attack’ response from the colony [23,48]. 

## 6. Chemical Properties of Alarm Pheromones

The behavioural response to the alarm pheromone (as observed in the receiver) may vary depending on the properties of the chemical signal. Chemicals are released in the air and diffuse away to reach a concentration below the sensitivity of the receiver. Thus, the communication of the signal and resulting behaviour of the receiver changes or comes to a halt. As for all chemicals, a higher quantity of pheromone (Q), or a lower threshold of sensitivity to receiver (K) increases the range of communication. This is expressed as the Q:K ratio. In addition, the diffusion constant (D) also influences the communication range: a chemical with high volatility (high D) travels far and fast, hence its spatial range is large but its temporal range is small. Concentration and diffusion properties of chemicals in the air or water have a large role to play, as they directly influence the behavioural output [47,79]. The alarm pheromone components are usually small in molecular size and have high volatility, so they can reach the receivers fast and are cleared quickly after the disturbance. For example, in *Pogonomyrmex*, the alarm pheromone remains ‘active’ for a time span of about 35 s, or a distance of about 6 cm [79], after which the ants do not act aggressively unless there is a constant release of the alarm pheromone. 

The spatial range of a pheromone is called its active space. Striking examples of active spaces can be observed from the response of some ant species to their MAP. The leaf-cutter ant *Atta texana* produces a single compound, 4-methyl-3-heptanone. The concentration gradient creates concentric rings around the releaser (see Figure 3). Nestmates situated in the outer ring, where the MAP concentration is low, are attracted towards the inner circle (site of attack) where the higher alarm pheromone concentration triggers a frenzy of attack against the intruder or alien object (Figure 3A). The same pattern can be produced by multiple chemical components differing in their volatilities [79] Indeed the action of the MAP from *Myrmicaria eumenoides* and from *Oecophylla longinoda* can be described by a number of concentric rings wherein different chemical compounds are dominant. Each compound elicits a specific behavioural response, adequately representing the distance to the threat (Figure 3B,C) [80,81,82]. The duration of the release of alarm pheromones also affects what kind of behaviour is triggered. *Pogonomyrmex* ants change their defensive behaviour to a digging behaviour, after smelling their alarm pheromone for a long time [79]. Similarly, alarm behaviour has been observed to sometimes decrease after repeated exposure to the alarm pheromone, as reported in a recent study on Argentine ants *Linepithema humile* [83]. Finally, the pheromone concentration can also provide non-spatial information. A recent study on honeybees shows that they become more likely to sting as the alarm pheromone concentration increases, however, this aggressive response drops back when very high concentrations of alarm pheromone are reached. This mechanism may help to prevent a disproportionate defensive response [84]. In another study, it was shown that the concentration of the alarm pheromone component varied between exotic species of fire ants (*Solenopsis richteri*, *S. invicta*, and their hybrid *S. richteri × S. invicta*) and the native *Solenopsis geminata*—thus suggesting a potential link between alarm pheromone production and invasion success [85]. 

## 7. Effects of the Alarm Pheromone in Different Behavioural Contexts

So far, we have discussed the releaser effects of alarm pheromone in defensive behaviour. The same alarm pheromone can sometimes have a long-lasting physiological effect, i.e., a ‘primer effect’ on recipient conspecifics and this can be explained with the example of the SAP in honeybees. After exposure to IAA (SAP compound), bees decrease their response threshold to nociceptive stimuli [86]. Furthermore, the behavioural response to IAA was affected for up to 2.5 h after a first exposure, and the gene *c-Jun* was selectively expressed in the honeybee brain after exposure to IAA and/or attacking behaviour [87]. *C-Jun* is an important transcription factor, hence its expression suggests long-term modifications in the function of these neurons. Other physiological changes such as increased alertness, withdrawal from non-defensive tasks [87] and a lateralization bias in orientation [88] have been observed to be a result of IAA exposure. Among carpenter ants *Camponotus aethiops*, exposure to the alarm pheromone, formic acid, induced a better olfactory discrimination of nestmates vs. non-nestmates [89]. Overall, these behavioural changes are appropriate for a defensive context, as they likely result in a more efficient defensive response. However, this comes at the detriment of behavioural performance in other contexts, for example, foraging. As mentioned earlier, Urlacher et al. [10] demonstrated this by showing that individual bees lose their ability to learn floral odours associated with a sucrose reward, after exposure to their SAP. Similarly, 2-Heptanone (MAP compound) modulates associative olfactory learning and memory retention in bees [90]. In the stingless bee *Tetragonisca angustula*, repeated exposure to its synthetic alarm pheromone components results in a memory decay and reduces components of the alarm response [91]. 

The context and internal state of individuals also affect their response to the alarm pheromone. For example, the presence of appetitive floral odours reduced the likelihood that honeybees respond to IAA by stinging [66]. The two different types of odours—one which is appetitive (floral) and another that signals alarm (IAA) here exert opposite actions that are in some way weighed and integrated into the bee brain. The state and size of the colony also affect defensive responses [67,92,93]. Internal factors such as genetic traits, early social life [94] and age/task allocation also change the response to the alarm pheromone in bees [95,96]. Differential responses to the alarm pheromone depending on age have also been reported in the ant *Platythyrea punctata* [97]. Overall, older workers tend to be more aggressive than young ones, probably because of their lower residual value to the colony. 

To conclude, multiple factors influence the reaction of an individual to the alarm pheromone, from the physical properties of the pheromone and the presence of other olfactory stimuli in the environment, to intrinsic factors such as genetic and colony traits, and then finally to the behaviour exhibited by other group members. Understanding how all these factors are encoded by the brain, and more specifically the role played by olfactory processing areas, is a challenging task for future research. 

## 8. Neurobiology of Olfactory Coding for Defensive Mechanisms in Insects

How are defensive mechanisms, such as detecting a predator, responding to the alarm pheromone by stinging, or attacking a non-nestmate by smelling cuticular hydrocarbons, sensed and processed? How is information sent to the brain, allowing the insect to select a specific behavioural task in defensive behaviour? In other words, how do odour molecules, pheromones either composed of single odorants or complex mixtures, evoke a particular behaviour in an insect? In this section, we give a brief overview of the physiology of insect olfaction. Next, we discuss examples of known neuronal processing, particularly focusing on alarm pheromones. We mention some of the gaps in the field that call for future research. 

Odorants in the environment are sensed by a pair of antennae on the head of the insect. Each antenna is supplied with muscles, which enables the insect to position or move the antennae in a way to maximize odorant detection. Cuticular structures called ‘sensilla’ are present on the surface of the antenna. Additionally, sensilla may be located on other organs, such as the maxillary palps, legs, wings, or genitalia–not all of these use the brain for further processing. Across species, there is a great variety of sensilla, differing in shape (hair-like, peg-like, placode-like, and several more), structure (single-walled, double-walled), neural size (from 2 to well over 50 neurons in a single sensillum) and neural composition (colocalized with mechanosensory neurons, with temperature-sensitive neurons, with humidity sensory neurons) [98,99]. Olfactory sensilla are filled with sensillar lymph, and innervated by the dendrites of olfactory sensory neurons (OSNs) [100]. Odorant molecules have to shift from being airborne to being dissolved in the liquid lymph, before interacting with the dendrites. Olfactory receptors, which are membrane-bound proteins residing in these dendrites, detect odorant molecules. Several types of receptors and accessory proteins are known in insects [101], the two most prominent families are the ORs (Olfactory Receptors *sensu strictu*) and the IRs (Ionotropic Receptors). Upon binding with a specific ligand, olfactory receptors lead to membrane depolarization in the OSN dendrite, generating action potentials that travel along the axons into the brain: insect OSNs are primary sensory cells that produce their own action potentials. 

OSN axons coalesce into antennal axonal tracts. These tracts then enter their first brain structure, the antennal lobes (Figure 4A). Here, the axon terminals end in glomerular structures. Generally, each glomerulus collects axons with the same physiological properties, i.e., axons from cells that express the same olfactory receptor type (or, in some cases, types). Thus, glomeruli represent the first functional units in the olfactory system. A dense network of local neurons (LNs) creates lateral information flow within each glomerulus, and across different glomeruli. Most projection neurons (PNs) are uniglomerular, i.e., their dendrites are located in a single glomerulus, where they collect information from OSNs and LNs–thus, information from these functional units is delivered to other brain areas [102]. Multi-glomerular PNs also exist, they have dendrites spanning a group, or sometimes all glomeruli: therefore, they have access to multiple functional units simultaneously. They also have output synapses within the antennal lobes, and therefore they also function as local neurons within the antennal lobes, interconnecting glomeruli. PNs send their axons to the mushroom bodies (notably uniglomerular PNs), and to the lateral protocerebrum (all PNs), along the antennal-protocerebral-tract (APT; in older literature, these tracts had a variety of names for different species). The APT has two main tracts and several smaller ones, forming a dual olfactory pathway connecting the antennal lobes to the rest of the brain [103,104,105]. This includes the medial and lateral antenna-protocerebral tracts (m-APT and l-APT), where information is transferred by uniglomerular PNs, and the intermediate medio-lateral tracts (ml-APT), dominated by multi-glomerular PNs. In the l-APT, PNs carry information from single glomeruli to the lateral protocerebrum and then to the dendritic arborizations in the mushroom body (MB) lobes, while the m-APT axons follow a reverse trajectory. m-APT axons have been stained with acetylcholinesterase, this suggests acetyl choline to be the main neurotransmitter in this olfactory pathway [106]. Similarly, GABAergic fibers are found along the ml-APT [107]. In addition to these feed-forward projections, information is also fed back by means of feedback neurons, thus creating a dynamic circuitry for olfactory information processing (Figure 4B). 

In the mushroom bodies, uniglomerular PN input from the antennal lobes provides odour information (mostly using the neurotransmitter acetyl choline), and modulatory input provides information about positive (e.g., food) or negative (e.g., predator exposure) events (using neurotransmitters such as octopamine and dopamine), thus enabling associative learning. The primary cells in the mushroom bodies are the Kenyon Cells (KC). In honeybees, there are about 180,000 KCs in each brain hemisphere: each PN synapses onto many KCs, and, conversely, each KC gets input from a particular pattern of PNs. Thus, each KC extracts a particular, combinatorial pattern of PN activity, creating a large multidimensional representation of odour quality. This is reminiscent of a support vector machine to computationally perform non-linear classification [108]. This architecture is ideally suited to categorize odours: it recognizes activity patterns across PNs that indicate a particular odour. The other target area of PN axons, the lateral protocerebrum, is not structured as geometrically as the mushroom bodies, making it more difficult to dissect the neural networks that are involved in odour processing. Generally, it is assumed that innate odour responses are generated in the lateral protocerebrum, and modulated by learned (plastic) odour evaluations from the mushroom bodies. Accordingly, within the lateral protocerebrum, there are dedicated areas for appetitive (positive) odours, and for negative (aversive) odours [109]. 

This anatomy of the brain supports two types of architectures for olfactory processing. The “labelled-line” architecture is particularly well suited for very specific and stereotypic olfactory responses. At the periphery, it consists of OSNs expressing a narrowly tuned OR, so that they are highly and exclusively sensitive to a particular compound. This creates a prominent glomerulus for this odour, and uniglomerular PNs can further carry the odour information to a dedicated circuitry in higher brain areas in order to generate a specific behavioural pattern. Since the labelled-line forms a direct link between an environmental cue and a behavioural output, it is always associated with odorants of high biological values such as food or host odours, and of course pheromones. Glomeruli involved in this type of olfactory processing may be enlarged, further reflecting the importance of this odour, in which case they are called “macroglomeruli”. However, such an architecture could not cope with the sheer number of chemical compounds that are relevant to an animal. A more efficient way to deal with the vast diversity of the chemical world is “across-fiber patterns”. In this alternative architecture, ORs tend to be broadly tuned, such that a given compound typically elicits responses in multiple glomeruli. The intensity of the response varies for each glomerulus, reflecting the different sensitivities of the corresponding ORs. Thus, the identification of an odour relies on its “signature” pattern and can only be computed by knowing the activity of the antennal lobe as a whole, which is the role of KCs as presented above [110].

In defensive behaviour, labelled-line systems of olfactory processing can be observed in both non-social and social insects. In *Drosophila*, flies under stress release high levels of CO_2_ due to their increased metabolic activity. High concentrations of CO_2_ elicit avoidance behaviour in naïve flies, mediated by highly selective OSNs sensing CO_2_ [111]. In another report, it was shown that the olfactory receptor Or49a in *Drosophila* is tuned to iridomyrmecin, an odour of its enemy (the *Leptopilina* parasitoid wasp). Activity in Or49a is carried on by PNs to the lateral protocerebrum, which behaviourally mediates the avoidance behaviour of adult flies [6]. Similarly, a recent study showed that specific odorant-binding proteins mediate the olfactory recognition of the alarm pheromone in the fire ant *Solenopsis invicta* [112]. Also, the “flee” response observed in aphids that drop from their host plant when detecting either the predator or the alarm pheromone from conspecifics is initiated by a single compound [52]. If this detection is accomplished by a dedicated OSN type, this would constitute a labelled line system. Hymenopterans like the carpenter ant *Camponotus obscuripes* have a more dynamic response to the alarm pheromone, which includes an initial alarm response followed by a stint of aggression towards the attacking predator. This response can be attributed to a labelled-line system at the level of PNs: recordings from uniglomerular PNs responding to the alarm pheromone show that the main neurite had dendritic arborizations in a specific group of glomeruli in the AL, termed ‘alarm-pheromone sensitive (AS) glomeruli’, and had axons in the MB lobes and lateral horn mediating the alarm response towards the predator [113]. Note that this is a labelled line that is not formed by a single OSN type, but rather by a pattern of glomerular activity, and therefore only visible at the level of central processing, in the PNs. Overall, labelled-line systems are useful to pass on the ‘key’ information to the animal, generally in an innate fashion, allowing for very quick and stable behavioural responses. When used in a defensive context, they may convey the direct information of predator detection, aversion, repulsion, and/or site-avoidance in non-social insects, communicate the single message of ‘drop’ from other conspecifics in some social insects, or control the initiation of alarm response by sending alarm message to the higher brain regions in Hymenopterans [114]. Apart from stereotypical and innate behavioural outputs, the second type of olfactory processing architecture, called “across-fibers” or “across-glomeruli” or “combinatorial” coding logic leads to a wide range of responses in the insect, including those that are more variable, and situation dependent. Here, the wide range of odorants bind to several odorant receptors, thus leading to activate multiple glomeruli in the antennal lobes of the brain. Calcium imaging studies have shown a characteristic species-specific ‘mosaic’ of activated glomeruli in *Apis mellifera* and *Camponotus rufipes*, in response to their respective alarm pheromones and other non-pheromonal odours [115,116,117]. In the ant brain, intracellular recordings and stainings showed results of an across-fibers pattern as well, i.e., multi-PNs had dendritic arborizations in both alarm sensitive glomeruli as well as other glomeruli, and gave rise to axons in the MB lobes terminating in the lateral horn, central complex and dorsal part of the protocerebrum [118]. In comparison to the uniglomerular PNs projecting to the MB lobes and providing direct information to control the alarm pheromone, it has been predicted that these across-fibers patterns have a role to play in the termination of ongoing activity and preparation of alarm response behaviour, by activating the sensorimotor system [118,119]. 

A detailed model of alarm pheromone processing was proposed based on the identification of 63 alarm pheromone-sensitive neurons in the ant brain [118,119] (Figure 5). The authors here suggest two parallel pathways depending upon the behavioural context, the first where a connection is observed from the sensory areas to the premotor and motor areas, via the lateral horn; and the second where the mushroom bodies also contribute to pheromone and general odour identification. The authors speculate that the first pathway is involved in alarm behaviour, where information from alarm-pheromone-sensitive glomeruli is transmitted to the higher brain regions, while the second is involved in aggressive behaviour, which responds to both alarm odour and other general odours. Identification of the Pe1 neuron responding to alarm pheromone in the ant brain is speculated to be homologous to the one identified in honeybees, responsible for short-term olfactory memory [120]. This neuron may be involved in the short-term retention of alarm pheromone signals. Furthermore, the identification of particular protocerebrum (PR) neurons in the ant brain, like the “widefield PR neuron” and the “alarm pheromone descending neuron” are suggested to be involved in multisensory integration of signals, decision-making processes, and orientation towards the pheromone in aggressive behaviour [118,119]. Although more work is required to characterize the physiology and anatomy of the neurons involved in alarm pheromone responses, these studies and the hypothetical functionality of alarm pheromone-sensitive neurons already give us an idea of how the neural correlates of alarm and aggressive behaviour may be realized.

Unfortunately, alarm-pheromone sensitive glomeruli have been only identified and characterized in *Camponotus* species, so far [118,119]. In honeybees, there has been no record of a specific set of glomeruli in the antennal lobe that processes alarm information in a non-ambiguous manner. Olfactory processing of pheromones and non-pheromonal odours are both represented in distributed patterns across glomeruli in the worker bee AL, and there is no indication of particular labelled-line subsystems. Calcium imaging studies have shown that chemical substances presented at ecological concentrations generally activate multiple glomeruli [121], and similarly, in the presentation of the components of the sting alarm pheromone, there was no identification of alarm-pheromone selective glomeruli [122]. Furthermore, all components of the sting alarm pheromone elicited a neural response in the glomeruli of the AL irrespective of their behavioural relevance [122], which is in line with another study that showed that the processing of the alarm pheromone was by a combination of broadly tuned glomeruli [123], instead of alarm-pheromone selective glomeruli like observed in *Camponotus* ant species. Other pheromones in *Apis mellifera*, like the queen and brood pheromones, showed differential activation in the m-APT and l-APT tracts in the brain, while social pheromones like the alarm pheromone induced a redundant activity in these pathways [123]. This confirms that alarm pheromone processing in worker honeybees is likely by means of a combinatorial or across-fibers coding system [115,116,117]. Thus, comparing across insect species, olfactory coding of alarm pheromones can either be represented as a labelled-line only system, an across-fibers only system, or a combination of both. 

Although, hypothetically, the olfactory processing may have evolved independently in two species of the same order Hymenoptera, it is also possible that a system of processing alarm pheromone across glomeruli, first, could have evolved at a subsequent stage into the dedicated olfactory pathway found among ants. From a theoretical point of view, the evolution of a dedicated labelled-line system for the processing of alarm pheromone also needs to be addressed. While the selective advantage of such a system is evident for the processing of ‘private’ pheromones, like sex pheromones as observed in moths, it is less clear in the case of alarm pheromones, which often also communicate across species, and may be more variable in evolutionary times. This is a question that needs to be explored with techniques that can access neural structures of projections in the brain at high resolution. Methods including electrophysiology, immunohistochemistry, and high-resolution optical imaging can help in understanding the alarm pheromone processing patterns and their diversities. Another approach for future studies is to compare the physiology of olfactory pathways between solitary and eusocial Hymenopterans, specific to their response to threat odorants and alarm pheromones. Such a comparative study, for example between solitary bees vs. honeybees or solitary wasps vs. social wasps, can provide a perspective of evolution in alarm pheromone processing. 

Aggressive behaviour is not only based on detecting appropriate threat stimuli, or pheromones, but also depends on humoral and modulatory responses. Biogenic amines including serotonin (5-HT), dopamine (DA), and octopamine (OA) are involved in the modulation of aggressive behaviour in invertebrates [95,124,125,126,127,128,129,130,131,132]. Also, cooperative defense behaviour is modulated by neurotransmitters like 5-HT and DA, as shown in honeybees [133]. Using high-performance liquid chromatography (HPLC) to determine the biogenic amine concentrations in different regions of the brain, and comparing bees exposed to the synthetic alarm pheromone (isoamyl acetate, IAA) to control bees [66], 5-HT titre values were shown to be increased in the central brain and suboesophageal zones, specifically. Moreover, bees with a shorter latency to sting had the highest 5-HT levels. DA levels in the central brain also increased upon exposure to the alarm pheromone [133]. This would propose central brain areas to be pivotal for aggressive behaviour, and likely the place where the olfactory information of the alarm pheromone is relayed to. It also opens another way to identifying the neural correlates of alarm pheromone processing in Hymenopterans: using immunostaining of these biogenic amines, the precise location of their upregulation could be monitored. Thus, future studies using immunostaining methods will not only help us characterize neurons and projections binding to specific neurotransmitters in the brain, but will also map out the pathways induced by the alarm pheromone. Combining with intracellular electrophysiology as observed in [134], where multimodal APT neurons showed an inhibited response to isoamyl acetate, will allow to determine the contribution of neurons to higher brain regions while traversing in the dual olfactory pathway. This can help us build a neural circuitry specific to defensive behaviour in insects. 

## 9. Outlook to Future Research

Defensive behaviour is both an important as well as a risky behaviour for any animal, at the edge of survival. Therefore, studying it has to consider several factors which may modify the decisions an individual makes in a defensive situation, ranging from its own inner state (age, hunger, reproductive status) to social cues (group defense, kin selection). So far, most studies have focused on defensive behaviour in insects relating to them being solitary or social species (use of chemical warfare or chemical communication), or on the type of threat faced, e.g., predator, intruder, or non-nestmate. Olfactory communication has been recognized as a major component, both for interspecific and for intraspecific information flow. 

In this review, we describe how odours make sense to an animal in an environment, the neurobiology of odour processing, and behavioural changes which are elicited by specific odours. However, in many instances, our knowledge is sparse: both in the sense that what is known in one species, has not been studied in another species, and in the sense that distinctive differences between concepts, such as alarm, defense, or aggression, still need to be integrated more extensively into experimental designs. Techniques like molecular biology and neurophysiology, immunohistochemistry and intracellular electrophysiology, paired with tangible behavioural assays will be of paramount importance to characterise neuronal processes underlying the coding of olfactory signals, from an anatomical and physiological aspect. The use of newer techniques like 2-photon microscopy, optogenetics, CRISPR transgenics, and more, can allow us to have a better perspective on the intersection of behaviour and neuroscience. Moreover, computational approaches, comprising mathematical modelling of alarm pheromone signalling in groups, and analysing behavioural information using video-tracking devices, can strengthen our understanding of olfactory interactions between individuals during a chemical signalling event such as a defensive behaviour bout. The study of animal behaviour and olfactory neuroscience, along with its interdisciplinary with computational science, can lead us to a network of understanding biological processes in a better way.

## Figures and Tables

**Figure 1 insects-13-00470-f001:**
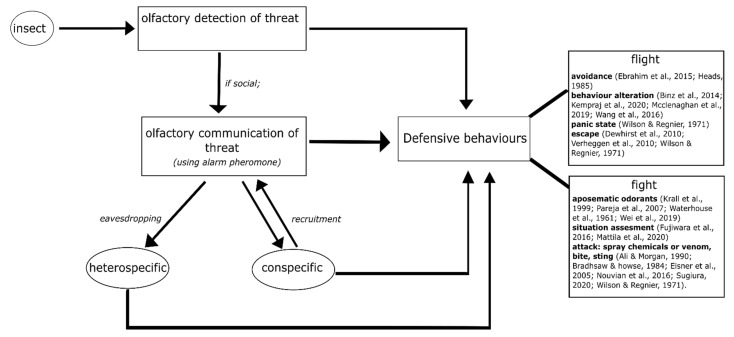
Olfactory strategies of insects in defensive behaviour: individual insects have a first strategy of olfactorily detecting their threat or disturbance and then eliciting a defensive behaviour. However, if they are members of a group and are social, the insects communicate the threat to their conspecifics by means of olfactory signals called ‘alarm pheromones’. Conspecifics respond to the alarm pheromone by eliciting a defensive behaviour and also by recruiting more conspecifics to perform the defensive behaviour. Sometimes, the alarm pheromone also acts as a signal to other species that share the same environment, i.e., heterospecific, leading them to a defensive behaviour against their common threat. Here, we show that defensive behaviour can mainly be classified into ‘flight’ or ‘fight’. ‘Flight’ includes avoidance of the predator/intruder, laying low-key to not attract attention, exhibiting a panic response by moving rapidly in circles or a zig-zag fashion, and escape. ‘Fight’ includes the use of aposematic or deterring odorants (along with a bitter taste) when attacked, masking with foul external odours to deter the intruder, and attacking–which ranges from biting, stinging, and spraying chemicals or venom. We have included key references next to the specific defensive behaviours, which are mentioned in detail in the main text and can be used for further reading.

**Figure 2 insects-13-00470-f002:**
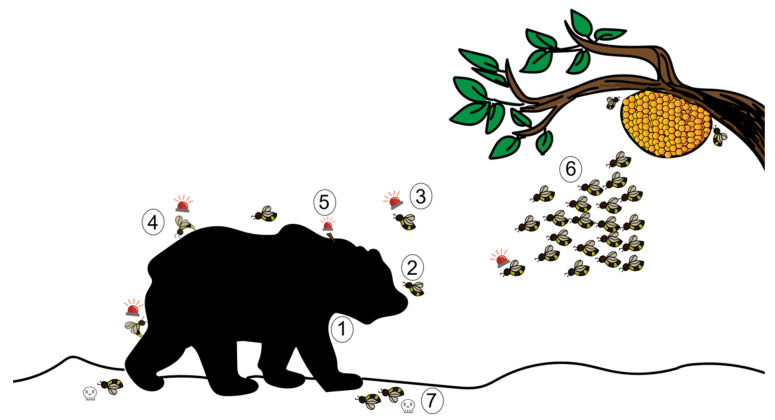
The defensive behaviour of a honeybee using sting alarm pheromone (SAP) against a mammalian predator. ①: A large mammal acts as a trigger to honeybees; ②: A guard bee flies in to check the threat; ③: Guard bees release alarm pheromones depending on the nature of the threat; ④: Recruited conspecifics react to the SAP by stinging the threat; ⑤: This results in the detachment of the stinger from the bee’s abdomen, which in turn also sends out SAP; ⑥: Groups of bees that are triggered upon release of the SAP attack the threat; ⑦: Stinging results in the death of the bees (an altruistic self-destructive act).

**Figure 3 insects-13-00470-f003:**
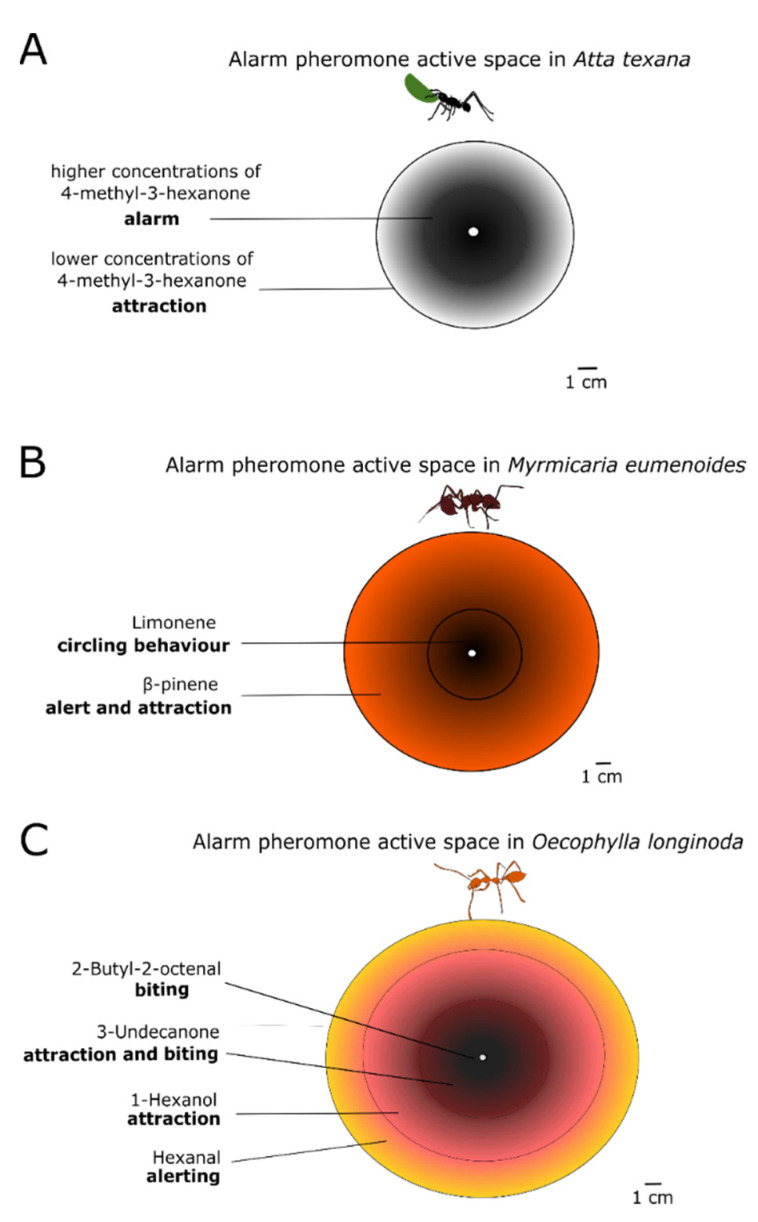
Different substances of the alarm pheromone or different concentrations of the same alarm pheromone substance form concentric rings around the releaser and are called ‘active spaces’. Here we see the pheromone active spaces created by three different species of ants. (**A**): The leaf-cutter ant *Atta texana* releases its mandibular alarm pheromone 4-methyl-3-heptanone in two different concentrations-outer ring of lower concentrations attracts nestmates to the inner concentric ring of higher which signals attack; (**B**): *Myrmicaria eumenoides* releases an alarm pheromone with two components, an outer ring of β-pinene that alerts and attracts nestmates and an inner ring of limonene that puts them in a circling behaviour of attack; (**C**): *Oecophylla longinoda* uses an active space of four concentric rings with different chemical compounds, which elicit a defensive behaviour from the outer to the inner, as depicted in the figure. Adapted from [82].

**Figure 4 insects-13-00470-f004:**
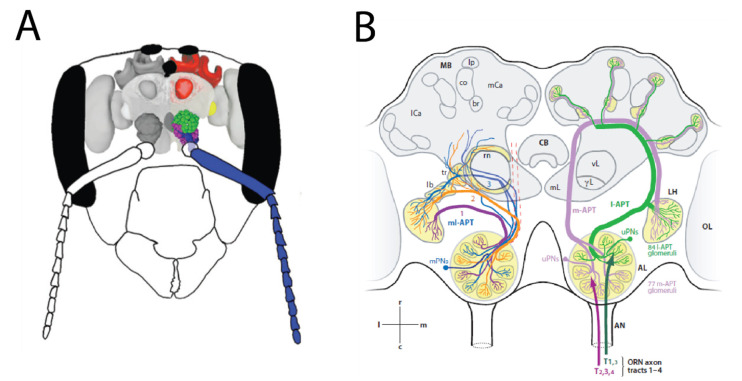
Olfactory coding in the honeybee neural system (**A**): Overview of the honeybee olfactory system in a schematic head capsule, with the main olfactory organs and areas (antenna (blue), antennal lobe (green and purple), lateral protocerebrum (yellow), mushroom body (red)), reprinted from [103] (**B**): Schematic overview of the dual olfactory system in honeybees, reproduced with permission from [104] 2006, John Wiley and Sons, Inc., For details, see text.

**Figure 5 insects-13-00470-f005:**
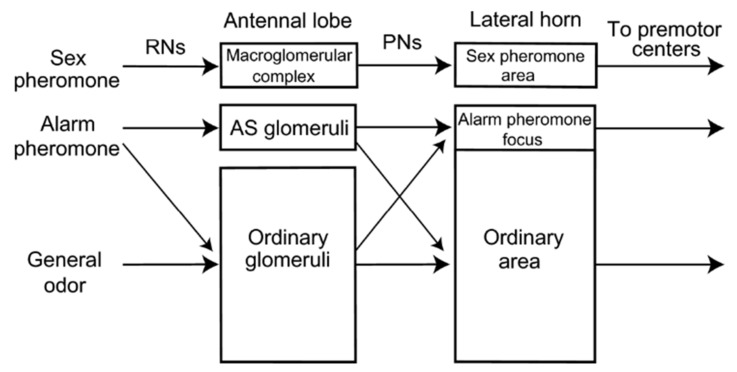
Model of alarm pheromone processing in the insect brain. Two parallel pathways are organised for the processing of odours in the insect brain. The first involves a direct path, called the “labelled line” architecture, where specific structures such as ‘alarm pheromone sensitive’ (AS) glomeruli as observed in the brain of *Camponotus* ant species are involved in processing to higher brain areas such as the lateral horn. This is compared to the processing of sex pheromones in male moths, which possess a specialised structure called the macroglomerular complex. The second, called the “across fibers” architecture is observed in species like *Apis mellifera* which do not possess specialised glomeruli for alarm pheromone processing. Here, the representation of the alarm pheromone is overlapped with general odours in the glomeruli and the lateral horn.. Reprinted with permission from [119], 2010, Prof. Makoto Mizunami.

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
