# Peer review of "Olfactory Strategies in the Defensive Behaviour of Insects"

_insects, 2022, doi:10.3390/insects13050470_

Round 1

Reviewer 1 Report

The authors define "defensive behavior" as behavior switching to survive an individual or colony and extensively review both behavioral and neurophysiological aspects of the olfactory system of insects. Although much remains to be studied about defensive behavior, many readers of this paper will find excellent information on insect olfaction in this review.
The main text is exceptionally well written and nearly complete enough for publication.

Minor comments:
1. The authors should fill in the blank "()" in LINE 177.

2. The aouthors should remove extra space between "the" and "air" in LINE 363.

3. The author should change "and" written in italics to normal in LINE 393.

Reviewer 2 Report

Dr. Galizia is one of the pioneer in the field of insect chemical ecology, and this manuscript from his lab (review article) will pave way to further study insect behavior related to defense mechanisms. The authors have illustrated very clearly how this mechanism to put to play in social and non-social insects. The major emphasis has been on the alarm pheromone in social insects, and further explain the behavioral and neural basis associated with this pheromone.

Having said that, I have very few concerns/ clarifications that can be easily addressed by the authors.

Figure 1: The figure is self-explanatory on different strategies insects deploy when encountered with a threat. However, the outcomes listed under flight and fight would require references (in parenthesis, maybe (?)) so that it would be easy for readers to follow up if these examples are further explained. If not discussed, references are handy for further reading.

L81: There is a fine line of difference when it comes to differentiating ‘laying low’ to avoid danger and crawling away from the predator.

L84-85: Does “similarly” refers to Tephritid fruit flies behavioral response to Leptopilina? The sentence doesn’t seem complete (?)!

L87: Freezing behavior is one of the examples of laying-low to avoid predation. The authors could re-arrange few sentences to present examples for “laying-low” and “espacing”. Reaction in fruit fly larvae to wasp sex pheromone is escaping behavior, while ‘freezing behavior’ in true fruit flies represent typically ‘laying-low’ to avoid predation.

L125&130: Typo

L159: “as we, humans, also avoid it for the same reason”: The context above is about the predation. I do not know the literature on human consumption of these bugs is ever published. We humans avoid touching them because they stink.

L177: typo

L415: Could you please include a brief explanation on what happens when the expression of c-Jun changes? What is the physiological role of this gene?

L422: Typo

L464- 584: This part is an important section of the manuscript, but not necessarily part of the defense behavior in insects. This section beautifully illustrate again (I say again because it has been shown on several occasions while explaining olfactory coding in different insect models) the mechanism of olfaction at the periphery and at the central nervous system. Yet, example of olfactory coding for moth sex pheromone component is not part of defensive behavior in moths (with due respect).

Reviewer 3 Report

Kannan et al. summarized the latest findings on the olfactory neural processing and defensive behavior in insects. The manuscripts is clearly written and provides an objective presentation of the current literature.

Specific points to be addressed are as follows.

L53 heterospecific —> plain font

L167 when —> When

L177 guard bees () : authors seemed to omitt some words. scientific name?

L250 There seem to be a space after the word “either”.

L250 fright —> flight?

L363 There seem to be a space before the word “air”.

L418 Authors write “withdrawal from other tasks“ two times. 

L527 180.000 —> 180,000?

L700 high-pass —> high-performance

Fig. 7 Spell down the abreviation AS to fig7 or to the legend.

Throughout manuscript

Please revise the citation below.

  • L422 (Urlacher, 2010) —> Urlacher (2010) 
  • L428 (Rossi et al., 2019) —> Rossi et al. (2019)
  • L434 (Nouvian et al., 2015) —> Nouvian et al. (2015)

Several citations are not listed in References. For example, Ono et al. 2003, Fortunato et al. 2001. Please check again.

Reviewer 4 Report

Dear authors.

Thank you for submitting this interesting and helpful review.

However the manuscript is still far from requirements for publication, needing an overall formatting and English writing review.

A review article should make a history of the subject, be based on main breakthrough old publications but also need to give a detailed “state of the art”; in the manuscript less than 20% references are within the last 10 years, which can barely be considered a relevant “state of the art”; even in social insects many recent publications are ignored (for instance a quick search in Insects there is a 2020 paper by Maccaro et al. on alarm pheromones in ants…

For instance the references along the text must be numbered and when several are mentioned, the sequence should be from old to more recent.

The reference list also need a  carefull review since has many mistakes and formatting errors; for instance: lines 768, 806, 811 have parts in capital letters  

Some more comments below

Lines 35-37 – Olfaction is predominantly used by insects in defensive behavior (Hansson & Stensmyr, 2011)

Line 54 – The different olfactory strategies of insects are described in detail (Fig. 1)

 Line 76 – missing species descriptor (Ischnura elegans) when first mentioned ???. Same in further species along the manuscript.

Line 96 – delete “this has been” replace by “also”

Line 109 – Vespa velutina in italic

Line 120 – replace situation by danger

Line 125 – space missing between words, the hive

Line 125 and 126 – remove sentence “The honeybee colony often succumbs to the attack.” Not relevant

Line 130 – space missing between words find the

Line 131 – remove “if they do, they”

Line 132 – remove “also”

 Line 135 – remove “. This includes” replace by “, such as”.

Lines 138-139 – “display forms of these defences that have been studied in detail “.

Line 144-145 -  “…pygidial glands that are strongly irritating to Humans and other animals.”.

Line 157 – “previously experienced predators.” Remove “with it”.

Line 159 – “…is a classic example of olfactory aposematism (Krall…)”. Remove all the rest of the sentence.

Line 177 – something is missing in “…guard bees () monitor…”

Lines 200-203 – “, so that an appropriate selective response can be elicited using various types of signals, such as, visual, acoustic and odours.”

Line 213 – “…by means of specific chemical compounds known as…”.

Lines 218-219 – don’t understand what authors mean with this sentence.

Line 226 – “has to be modulated by individuals’ condition “

Line 270 – “Formica species, …” species must not be in italic.

Lines 278-288 - Merge experiences descriptions.

Figure 2 caption – missing source (adapted from….).

Line 306-307 – “…use the alarm pheromone secreted from the mandibular glands…”.

Line 333 – “ … possibly apart from the defensive action against enemies, was later used as a trigger…”

Line 344 – “-each triggering different behavioural output on the receiver.”.

Although this action is not exclusive for alarm pheromones (same in aggregation or sexual pheromones, and that’s why these specific chemical compounds are largely used as lure placed in traps for the Integrated Pest Management. Some reference should be made.

Line 393 – reference (Wang, Wen, et al., 2016) is incorrectly written.

Lines 457 – “Nouvian et al. (2015) described…”.

Lines 546-584 – These In a review about defensive behaviour of insects I don’t understand this description on a moth sexual pheromone process. Why not include also the complex attack procedure of forest bark beetles mediated by aggregation and sexual pheromone communication and also involving fungi cooperation and decoding by secondary insects such as cerambycids that rely on tree hosts under attack to complete their life cycles?

Either remove or this part inclusion need to be better justified .

Line 664 – “…with a recent study…”. Recent is 2015. This brings to another global issue in the manuscript

Line 729 – “In this review is decribed how…”

Line 731 – delete “which have all been studied in some detail.” Obvious or it would be mentioned…

Line 738 – “… to characterize neuronal processes underlying the olfactory signals coding…”

Lines 738-739 – remove “Advancements in technology, with”. Start the sentence “the use…”
